# Experimentally Validated Finite Element Analysis of Thoracic Spine Compression Fractures in a Porcine Model

**DOI:** 10.3390/bioengineering11010096

**Published:** 2024-01-18

**Authors:** Sacha Guitteny, Cadence F. Lee, Farid Amirouche

**Affiliations:** 1Department of Orthopaedic Surgery, University of Illinois College of Medicine at Chicago, Chicago, IL 60607, USA; sacha.guitteny@univ-eiffel.fr (S.G.); clee331@uic.edu (C.F.L.); 2Orthopaedic and Spine Institute, NorthShore University Health System, Chicago, IL 60611, USA

**Keywords:** finite element, vertebral compression fracture, porcine, wedge fracture

## Abstract

Vertebral compression fractures (VCFs) occur in 1 to 1.5 million patients in the US each year and are associated with pain, disability, altered pulmonary function, secondary vertebral fracture, and increased mortality risk. A better understanding of VCFs and their management requires preclinical models that are both biomechanically analogous and accessible. We conducted a study using twelve spinal vertebrae (T12–T14) from porcine specimens. We created mathematical simulations of vertebral compression fractures (VCFs) using CT scans for reconstructing native anatomy and validated the results by conducting physical axial compression experiments. The simulations accurately predicted the behavior of the physical compressions. The coefficient of determination for stiffness was 0.71, the strength correlation was 0.88, and the failure of the vertebral bodies included vertical splitting on the lateral sides or horizontal separation in the anterior wall. This finite element method has important implications for the preventative, prognostic, and therapeutic management of VCFs. This study also supports the use of porcine specimens in orthopedic biomechanical research.

## 1. Introduction

Vertebral compression fractures (VCFs) occur in roughly 1–1.5 million people in the United States annually, with 60–75% of those fractures occurring in the thoracolumbar region (T12–L2) [1]. VCFs are classified as wedge, biconcave, or crush fractures, with wedge fractures comprising over 50% of all incidents [2]. This flexion fracture pattern, defined as an anterior collapse of the vertebra and intact posterior column, results in the characteristic “wedge”-shaped deformity [3]. The transitional zone between the relatively static thoracic spine into the mobile lumbar spine is vulnerable to biomechanical stress, explaining the large proportion of vertebral fractures in the thoracolumbar region. Patients with VCFs experience significant impacts on morbidity and mortality, including pain, disability, altered pulmonary and respiratory function, secondary vertebral fracture, and 72% and 90% mortality risk at 5 and 7 years following fracture incidence, respectively [4]. Current clinical management of VCFs includes pain management, physical therapy, rehabilitation, and, if necessary, surgical intervention by vertebroplasty or balloon kyphoplasty when the Thoracolumbar Injury Classification and Severity (TLICS) scale score exceeds a value of 4. Vertebroplasty is a minimally invasive procedure that stabilizes the fractured vertebra by injecting the fracture site with polymethylmethacrylate (PMMA) [5]. Contrast-enhanced fluoroscopy is used to guide the bone biopsy needle into the proper positioning to inject PMMA into the fractured site [6]. Kyphoplasty involves the insertion of an inflatable balloon tamp, creating a space to later fill with PMMA. Kyphoplasty is useful in restoring height and improving symptoms related to kyphosis. There is debate as to whether vertebroplasty or balloon kyphoplasty produces better outcomes, but both procedures are indicated for the surgical management of VCFs [7]. Despite their overall efficacy, these surgical techniques involve adverse complications, such as bone fragment retropulsion, PMMA leakage into the spinal cord (which may result in neurologic complications), and increased incidence of adjacent VCFs, due to the sharp contrast of material density [8,9,10]. To improve the clinical outcomes following vertebroplasty and balloon kyphoplasty, surgeons may benefit from more accurate surgical planning with 3D-simulated models that accurately depict the spinal environment. Additionally, a better understanding of the spine’s response to load and initiation of compression fractures may lead to preventative innovations, even prior to surgical intervention. Here, we seek to validate a mathematical finite element method (FEM) to compare with an experimental physical porcine vertebral fracture study in order to understand VCF biomechanics better. The numerical model and the experimental studies are specifically intended to replicate the vertebral compression “wedge” fractures to validate existing work in this area and further support the porcine specimen as an analogous model for spinal research.

The finite element method, or finite element analysis (FEA), is a computer-based mathematical method of analyzing the behavior of a given structure or material by dividing the body into small boundaries or finite elements and applying constraints to simulate unknown qualities [11,12]. The history of FEM can be traced back to 1851, from the concept of deriving differential equations of the surface area of an enclosed, irregular space, which utilizes discretizing the body into smaller triangular elements [13]. Numerous insights followed throughout the 1900s, utilizing FEM as a tool to evaluate stress and elastic behavior of materials in relation to external forces, such as aircraft wings during World War II [14]. Naturally, FEA became a popular tool in engineering, from aeronautical simulation of aircraft behavior to architectural dissection of historic buildings [15,16]. Recently, FEM has been applied in a biomedical context to evaluate medical pathologies, such as fluid dynamics in cardiovascular medicine, dental implants, hip arthroplasty, and cervical disc degeneration [17,18,19,20]. FEM is a valuable tool for determining the body’s physical response to different pathologies as well as the modification of implants and augmentative procedures on native structures and biomechanics.

The use of the FEM to emulate vertebral body fractures is well established [21,22,23], but prior studies have noted the potential for optimizing these predictive models, particularly in the thoracolumbar region [24]. One such area for improvement is the use of the follower load, which prevents rotations and shear forces in the intervertebral joints during axial (craniocaudal) loading. This method also has limitations in inducing additional unbalanced loading, as the forces follow the follower position and connect to other vertebrae, limiting the individual responses of connective tissues and ligaments. In this study, we incorporate follower load and support previous work [25,26,27,28] using a modified procedure to validate the QCT-based FE models for evaluating wedge-shaped VCFs.

We utilize computed tomography (CT)-scan-derived 3D models of vertebral bone and perform FEM to determine biomechanical patterns, such as fracture lines, stress–strain behavior, and responses to load. The concordance of the physical compression experiments and FEM will give support of the CT-derived numerical model’s ability to predict VCF behavior for incidences in which physical response cannot be validated, such as in a clinical patient setting.

Our use of porcine specimens is motivated by the anatomical concordance of human and porcine spines. Previous studies have examined the porcine spine as an appropriate comparison to the human’s and found analogous bone composition, density, and morphology [29,30]. Comparisons of the thoracic region include vertebral body width of the porcine spine at 50–80% of that in humans, depth of 50%, height of 90–120%, and 14 thoracolumbar vertebrae in porcine spines compared to 12 in the thoracolumbar region in humans [31]. As for mechanics, a study found that the range of motion of porcine spines is qualitatively similar to that of humans under various loading directions [32]. Overall, numerous studies have supported the porcine model as one of the most representative animal models for spinal research. This comparison is significant because animal spines are more cost-effective, readily available, and provide an advantage over the age constraints of typical human cadaver specimens. Since the study of human specimens is limited to the use of cadavers, much of the existing research on physical compression is restricted to the study of elderly, osteoporotic vertebrae, rather than younger, healthy specimens such as those from adolescents. Therefore, our aim was not only to validate a model of VCF with an improved protocol of FEM but also to assess the usefulness of porcine specimens in spinal research. This work’s foundation will enable further exploration of porcine–human comparisons in VCF.

## 2. Materials and Methods

### 2.1. Sample Selection and Preparation

VCFs are extensively described for the thoracolumbar region, as most spine fractures occur in the thoracolumbar area [33,34]. The vulnerability of this specific region is thought to be attributed to the transition between the rigid thoracic region and the flexible lumbar region [2]. Anatomically, the center of gravity of the spinal column is anteriorly weighted, such that compressive forces are loaded on the vertebral bodies, while the posterior vertebral elements, ligaments, and paraspinal muscles are loaded in tension [35]. Prior evaluation has shown that the thoracolumbar region is particularly vulnerable to these compressive loads when undergoing axial forces, and the mid-thoracic vertebral bodies may be further susceptible to failure due to their strain distribution from cancellous and cortical bone [36]. Due to the biomechanical vulnerability of this region and ventral-weighted susceptibility, we chose the lower thoracic vertebrae (T12–14) for studying wedge fractures. The morphological and qualitative similarities between porcine and human spines are well established and motivate our use of porcine specimens. Although the porcine spine has 14 thoracic vertebrae compared to 12 in humans, previous evaluation has shown notable anatomical similarity and mechanical comparability of the porcine thoracic spine under various loading directions [31,32]. Using the porcine spine provides data to support our FE model and future comparative studies.

The workflow of this study entails two processes. The first process is the physical axial compression of the vertebral bodies and evaluating their response to compressive load. The second process is the finite element simulation of axial compression based on CT-derived models of the vertebral bodies. The results of the physical compression alongside the FEA will allow us to make comparisons and assess the accuracy of the FEA in determining real fracture patterns and failure response.

We obtained four adolescent porcine spines from Peoria Packing Butcher Shop, Chicago, IL, containing twelve intact thoracolumbar vertebrae. The spine elements were grossly dissected in the UIC anatomy laboratory (Figure 1). Each vertebra was cleaned of muscles, ligaments, intervertebral discs, and any soft tissue. Posterior elements were removed, and endplates were partially removed. Finally, individual vertebrae were polished to ensure a planar surface for the axial compression test. The polishing of the endplates entailed using a scalpel to finely remove any remaining connective tissue or protruding bone that may alter the transverse plane of the compression experiments. This was carried out to achieve an appropriate planar surface in order to apply a proper axial force and simulate anterior wedge fracture.

### 2.2. CT and Numerical Modeling

Twelve individual vertebral bodies were subjected to computed tomography (CT) imaging (Table 1) in order to generate FE meshes.

Next, we utilized MIMICS v2.0, software that incorporates reference points from medical imaging techniques, such as CT, MRI, and ultrasound, to create 3D reconstructions of anatomical objects [37]. MIMICS software was utilized using the threshold method to build a 3D surface for each vertebra, derived from the CT images previously described. These 3D reconstructions were imported to 3-MATIC to refine the vertebrae and define relative planes (Figure 2). 3-MATIC v15.0 is software used in partnership with MIMICS, which takes the 3D reconstruction and creates a “mesh”, by triangulated segmentation of the organic surface [38]. The MIMICS-3-MATIC combination is well established in computer-aided design (CAD) for a range of finite element analyses but especially for spine research [39,40,41]. The average of the cranial and caudal surfaces was used to establish the vertebral transverse plane. This transverse plane identifies the vertebral body’s two endplates that can be cut to form the final upper and lower plates. The sagittal plane is perpendicular to the transverse plane, passing through the center of mass of the vertebra and following the spinous process. Posterior vertebrae elements were cut according to the sagittal plane, following the pedicle’s curvature. The corresponding 3-MATIC mesh comprises finite tetrahedral elements T4 with a defined edge length of approximately 1 mm. With MIMICS, the material is assigned to the volume mesh.

Next, mineral calibration equations were developed to convert CT grayscale values to density values. Holding CT specifications constant, we found the conversion between the CT Hounsfield Unit and the physical density, as previously described by Silva et al. [42]. Briefly, we performed simple density tests on 12 cortical cores from the lower and upper facets of the porcine vertebra (Figure 3) and assumed linear equations (Equation (1)).

The CT minimal grayscale value limited the density to 0.01 g/cm^3^ to avoid nonphysical negative density values. Thirty cores were analyzed to describe the whole material properties of the vertebral body. Constants a and b were calculated according to Equation (1), where ρ represents the density and HU is the abbreviation for Hounsfield Units:(1)ρ=a+HU×b

The material assignment in this study accounted for the anisotropy and the elastic damage in the vertebral bone. The material assigned to each finite element was assumed to be isotropic. Equation (2) was computed in MIMICS, defining the Young’s modulus (*E*), the yield strain (*ε_y_*), and the Poisson ratio (*ν*), as previously described [43]. The Young modulus was calculated as a function of the apparent density and assumed ratio *ρ*_app_/*ρ*_ash_ equal to 0.6.
(2)$$E=3050\rho^{1.81}\;\varepsilon_{y}=0.0065\rho^{-1.42}\;\nu =0.3\;$$

The 3D model and mesh were exported in an INP file to the ABAQUS software (version 2020) and edited using MATLAB code to add plastic behavior, as previously described [44]. For each finite element, the post-yield behavior was assumed to be perfectly plastic, described by Equation (3), with constraint (σ):(3)$$\sigma =E\varepsilon \;if\;\sigma <\sigma_{y}\;\sigma_{y}\;if\;\sigma \;\geq \;\sigma_{y}\;$$

The resulting vertebral body with its refined mesh and material assignment are represented in Figure 4 and Table 2:

### 2.3. Finite Element Analysis (FEA)

FEA was run on ABAQUS software for all T12–T14 vertebrae in one porcine spine. Using static and nonlinear analysis, the vertebral model was constrained to pure axial compression. The controlled point location was found using the follower load technique, where the follower load was tangent to the spine’s curve so that the controlled point was located at 10% of the vertebral body width along the sagittal direction from the center of mass projection in the upper plate (Figure 5). This technique ensures the production of the desired anterior wedge fracture. Multiple Points Constraints (MPC) were used to link the controlled point with the nodes constituting the vertebra’s cranial surface. The use of MPCs entails selecting nodes within the surface that will be constrained and assigned a zero-displacement value, fixing the surface. The MPCs ensure that there is no over-constraint of the surface. The ABAQUS tool created analytical links between the issues and the nodes. The transverse displacements of the loading point were locked, and the three rotations of the degree of freedom were left free, while all degrees of freedom of the caudal surface were locked.

Post-processing analysis of load–displacement and stress–strain characteristics involved the identification of the strength and stiffness of the vertebral bodies. Stiffness was defined as the slope in its linear trajectory, and strength was defined as the highest point of elastic response. Finally, the failure pattern was determined by identifying the fracture along the compression load from finite elements, with a nonzero plastic strain.

### 2.4. Experimental Testing

#### 2.4.1. Apparatus and Configuration of the Vertebrae in the MTS Machine

A specially designed MTS tensile machine was configured explicitly for experimental compression of vertebral bodies. MTS calibration was performed in compliance with accredited ISO/IEC 17025 [45] testing and calibration laboratories, and initial setup was validated prior to data collection. The upper loading plate was loosely adjusted to the MTS connector to allow for free rotation of up to 5°. The vertebral body’s lower profile was traced onto the lower compression plate to ensure proper placement of the specimen, such that the center of loading aligned with 10% of the width of the vertebral body from the center of mass in the sagittal direction. We created 3D plastic polymer disks of the vertebral body’s lower profile with the CT-derived numerical models. A laser cutter machine used four holes to define sagittal and coronal planes through the center of mass. Reference points and 3D reconstructions correctly positioned the vertebral bodies (Figure 6). The specimens were then subjected to a pure axial load of a quasi-static displacement rate of 3 mm/min, where failure was estimated as ⅓ of the vertebra height.

#### 2.4.2. Data Collection and Measurements

The MTS cells collected load and extension values to compute the load–displacement curve. OPTOTRAK equipment tracked motion changes on the upper surface of the vertebra, utilizing a digital marker fixed on the loading plate (Figure 7). Like the MTS calibration, OPTOTRAK equipment was validated with reference frames prior to data collection. The displacement of the center marker was calculated using the coordinates of three digitized features on the scale. The rotation of the upper surface was calculated using four digitized tags of the sagittal and coronal planes. A standard iPhone was also used to collect videos of compression experiments and pre- and post-compression photos of the vertebral bodies in all views.

## 3. Results

### 3.1. Experimental Compression

Vertebral specimens were derived from four porcine spines (assigned identifiers: P1–P4) for the thoracic vertebrae from T12 to T14. The pattern of load–displacement curves was similar across all models (Figure 8). Average vertebral stiffness was calculated as 9.54 ± 1.1 kN/mm (range 6.73–13.63 kN/mm), and average vertebral strength was 10.2 ± 0.86 kN (range 8.1–13 kN). The resulting stiffness and peak load values were grouped according to the vertebra level, T12–T14 (Figure 9). We did not notice crucial differences between the fracture response between vertebral levels. The average error in the stiffness and strength value was 24% and 17% in the porcine spine, respectively. Regarding the failure pattern, specimens displayed a transverse fracture line in the anterior wall, causing the superior and inferior anterior cortical shells to peel. The resulting maximal tilt during the compression in the sagittal plane was 1.25° for the porcine vertebral bodies.

### 3.2. Video Imaging Analysis

During the experimental compression tests, video analysis revealed a consistent failure pattern in all specimens. The pattern comprised three fracture lines; the first two were visible in the posterolateral parts of the vertebral body, where fractures propagated vertically from the superior to the inferior surface. The third fracture pattern was a midline split in the transverse plane, which occurred superiorly in the vertebral body, depending on the geometry of the anterior wall. This resulted in a peeling pattern in the superior and inferior anterior walls. Fracture propagation in oblique directions produced triangular lines in the anterior wall, creating the desired anterior wedge fractures. The fracture separated the anterior cortical shell from the top and bottom surfaces. Figure 10 shows the intact and fractured T14 vertebral body of porcine #4 in the anterior and lateral views, representing the failure pattern.

### 3.3. FEA Analysis

We performed a simple statistical analysis of the correlation between the stiffness and strength values recovered numerically and experimentally, including the error percentage. First, the results given by the FEA were sorted. We observed that the prediction of the stiffness of the porcine bodies was sufficient, showing an average error of 9.9% (range 1.3–18.4%). However, the recovered strength was low compared to the one from the experimental tests (average of 67% of error).

Because the errors on the peak load for the porcine bodies were similar for each vertebral level, we decided to impose a scaling factor on the yield strain calculation equation, which we found was equal to 3.1. This modification produced a resulting stiffness and strength from numerical analyses for the porcine vertebrae of 9.6 ± 0.9 kN/mm (range 8.4–10.5 kN/mm) and 9.9 ± 0.5 kN (range 9.6–10.7 kN), respectively. This scaling increased the average stiffness error to 28% (range 17–42%) but significantly improved peak load prediction, with an absolute average error of 2.8%. Furthermore, the resulting scaled FEA prediction correlated with the experimental observations of load displacement (Figure 11). A high coefficient of determination of 0.71, *p* < 0.01, and 0.88, *p* < 0.01 for the stiffness and the strength indicates a relevant correlation between our modeling and the experimental data.

Visually, FEA showed a similar pattern of splitting as observed in the physical compression experiments (Figure 12). As the load increased, the computer-generated vertebrae displayed an increase in strain on the anterior and lateral surface of the vertebral body, as determined by the plastic strain equivalent (PEEQ). The failure primarily occurred in the midtransverse plane, with bilateral failure of the vertebral body, except for sparing of a midline superoanterior ellipse. As the load progressed, the increasing strain was observed on the midtransverse anterior surface, as well as on the anterior aspect of the superior endplate and subsequently on the posterolateral zones of the superior endplate. From a superior view of the top surface, strain was seen throughout the circumference of the vertebral body but it was most prominent in the anterior and posterolateral zones. The distribution of load towards the anterior aspect of the vertebrae was expected due to our desired simulation of an anterior wedge fracture. From this superior view, the areas of strain appeared to correspond with the cortical shell of the body, sparing the area of central trabecular bone. By visualizing the progression of strain throughout the vertebral body, we can gain insight into the failure pattern observed experimentally, with fracture lines propagating from the cranial to caudal surface in the lateral and anterior sides.

## 4. Discussion

This study aimed to produce a CT-based FE model by experimentally simulating wedge fractures in porcine thoracolumbar vertebrae and support the use of porcine spines in human comparative studies. This study utilized 12 porcine experimental axial (craniocaudal) compression tests, including a numerical model for vertebrae T12–T14 of one spine. Initial comparisons between the experimental compression tests and the FEM numerical prediction indicated a low margin of error for stiffness (9.9%) but a relatively wide error in strength prediction (67%). Therefore, we utilized a scaling factor on the yield strain calculation to modify the predicted values and resultant errors for stiffness (28%) and strength (2.8%). In all experimental compression tests under the MTS machine, the twelve vertebrae showed a consistent pattern of results, as shown in Figure 8. We notably utilized the follower load technique to more accurately simulate axial compressive load, considering the curvature of the native spine. The consistency across experimental compression tests allowed us to run FEA on the vertebrae of just one porcine spine. The failure pattern of the vertebrae involved a transverse fracture in the anterior zone and fractures propagating vertically from the cranial to the caudal surfaces in the lateral zones. After closer examination of the porcine vertebrae, we noticed that the bone did not completely join with the vertebral body but enabled the junction with the transverse process on the lateral posterior side. This may be a result of a discrepancy between the porcine and human vertebra, which may be evaluated in future studies.

With FEM, we were able to determine the pattern of progressive failure of the experimental compressions. The origin of failure in the midtransverse plane suggests a potential failure of the trabecular bone rather than cortical bone endplates, resulting in the splitting of the lateral surfaces. Several studies have evaluated the interaction of cortical and trabecular load sharing in vertebral bodies but failed to address the role of bone composition in midtransverse thoracolumbar wedge fractures [46,47]. For clinical interest, it is important to elucidate whether the results we see here are a result of the use of a porcine model or a result of using adolescent, non-osteoporotic specimens.

It is important to investigate the concordance between porcine and humans for the outcomes of this study. Additionally, some amendments need to be made to overcome the existing limitations. Firstly, it is necessary to obtain a more accurate representation of the bone material and microstructure across the vertebral body. In our equations, we performed what might be perceived as simple and accurate calculations for density using mass and volume, which may not accurately represent the nonlinear material quality. It is particularly important to have an accurate depiction of these material properties when modeling VCF, considering the relationship of cortical and trabecular bone within the vertebral body. This requires an in-depth morphological description beyond the scope of this project. As previously described, the FEM pattern of strain in areas of the midtransverse body followed by the subsequent strain of the circumference of the cortical endplate suggest an importance of the load sharing properties across all areas of the vertebra. Therefore, more detailed imaging techniques such as micro CTs that have been used in prior VCF evaluation may lead to enhanced results and a more accurate simulation of the bone microstructure [48]. Additionally, the constraints of current imaging technology confined our observation of physical compression to external fractures rather than evaluating patterns seen within the interior of the vertebra. It would be helpful to know if failure was initiated internally or if any other gross changes were observed within the trabecular bone. A second limitation of this study is the plate attachment for the physical compressions. The mechanical experiments would benefit from incorporating a ball joint to allow for rotation of the loading plate, as described by Dall’Ara et al. [28]. The present study utilized a loose plate attachment, which allowed for limited tilt and rotation of less than 7°. Our MTS machine apparatus has a design limitation and may require a controlled environment with induced rotation to replicate real spine motion such as that seen in extension and flexion. A third consideration for future study is the use of alternative spine models or regions. Several studies have noted that the porcine cervical spine may be a more analogous model for the human lumbar spine as it relates to anatomical, geometric, and functional comparison [49,50]. Future studies may elucidate the impact of this concept and whether cervical or thoracolumbar vertebrae serve as a more accurate model of human thoracolumbar VCF. Furthermore, the evaluation of vertebral wedge fractures is clinically relevant for humans given the bipedal gait and gravitational weight distribution across the vertebral bodies, which is not an analogous biomechanical concern in the quadrupedal porcine spine. Despite numerous studies outlining the similarities between human and porcine spinal anatomy and range of motion, the altered weight distribution on the spine, increased axial length, and altered bone density are clear discrepancies in modeling and may present a limitation. Finally, future studies would benefit from evaluating sex and age differences between specimens. Male versus female anatomy maybe be of interest, as sex-based differences play a role in bone density and osteoporosis, as well as the incidence of VCF. Assessing VCF risk with increased age is also significant, as clinical trends suggest that the age of vertebral structures should be considered when evaluating fracture models [14,15]. The present study uses young, adolescent porcine of undetermined sex, which may alter the clinical utility of these numerical predictions.

## 5. Conclusions

Patients experiencing VCFs suffer from pain, decreased quality of life, and increased mortality risk. Current treatment of VCFs includes potential surgical intervention with or without rehabilitation devices, such as thoracic, lumbar sacral orthosis (TLSO), hard or soft cervical collars, and sacral corsets, which may be uncomfortable, decrease quality of life, and have low adherence. To better understand the mechanics and behavior of the spine under axial compression in VCFs, we created a finite element model to simulate failure patterns. Our proposed finite element model utilizes thoracic vertebral bone strength, stiffness, fraction pattern, and fracture location through CT scans of porcine spines to create a predictor of wedge compression fractures. The results from the experimental compression supported the accuracy of our numerical finite element model. We found that fracture patterns in porcine vertebrae were in the anterior and the posterior lateral areas, with failure initiating in the midtransverse body and subsequent superior endplate, causing lateral splitting.

In summary, the FEM was successful in accurately predicting the load–displacement observed in the physical compression experiments. Our findings complement similar studies that have investigated finite element models on the human spine but also support the use of the porcine model in comparative studies. Animal specimens such as porcine samples overcome some of the limitations of human cadaver studies, such as high cost, limited accessibility, and a narrow range of demographics; notably, human cadaver selection entails a lack of young, non-osteoporotic samples. Additionally, this modeling approach enhances previous work by implementing the follower load technique and can be used for future studies examining vertebral compression fractures and their risks, impact, treatment, and prevention in a clinical setting. A better understanding of wedge fracture biomechanics will contribute towards improved surgical management by identifying the areas that require reinforcement without compromising subsequent vertebral stability. CT-based models are patient-specific and, if complemented with more accurate morphology of the bone and material properties, can be extremely useful in surgical pre-planning. By performing a large number of simulations under different scenarios and loading conditions, we may contribute to a universally shared database so that these models can be standardized for testing.

## Figures and Tables

**Figure 1 bioengineering-11-00096-f001:**
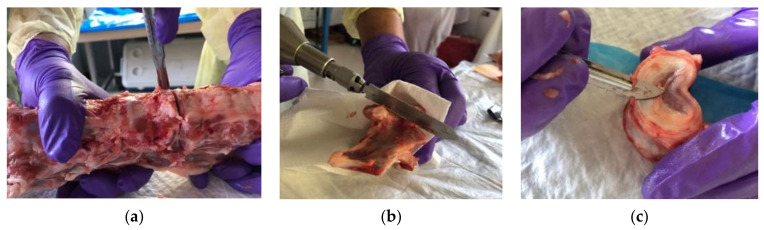
Specimen preparation from four adolescent porcine spines, comprising twelve thoracolumbar vertebrae. (**a**) Separation of vertebrae within each spine, (**b**) removal of posterior elements including pedicle, spinous process, and transverse process, (**c**) dissection and removal of soft tissue, finely polishing the endplate.

**Figure 2 bioengineering-11-00096-f002:**
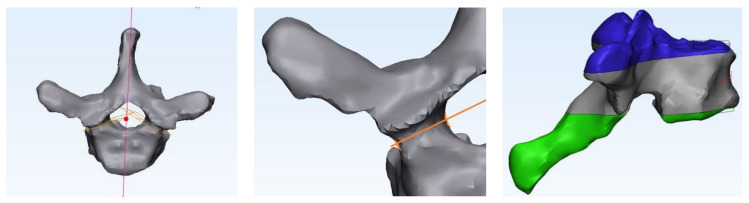
Vertebral planes defined for each 3D reconstruction from MIMICS and 3-MATIC, distinguishing the vertebral body from posterior elements and establishing the transverse planes.

**Figure 3 bioengineering-11-00096-f003:**
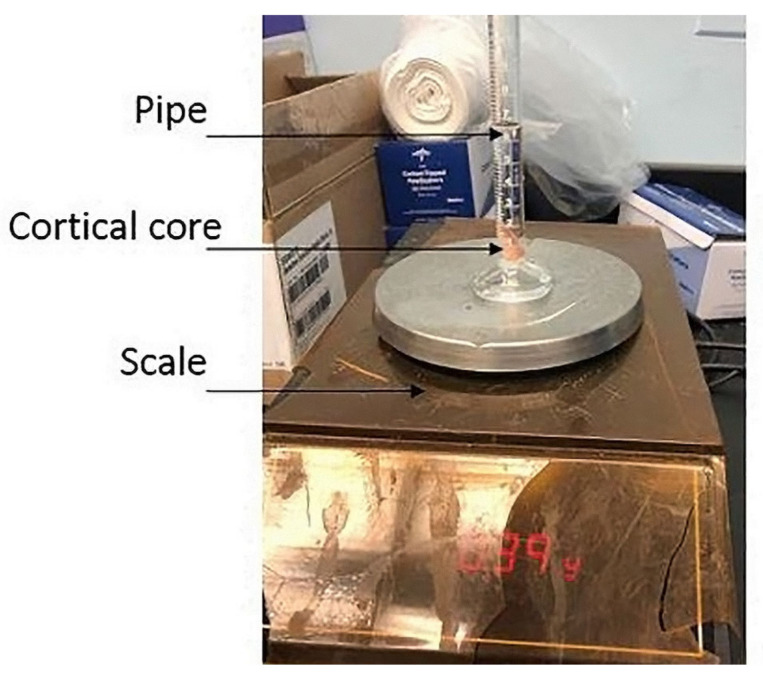
Density test weight and volume scaling of a vertebral cortical core.

**Figure 4 bioengineering-11-00096-f004:**
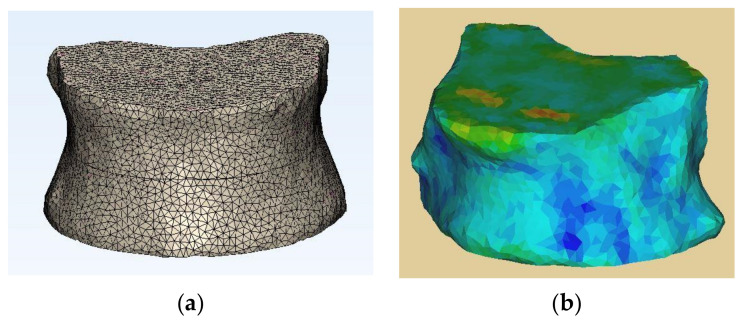
An example of a vertebral body FE model with its (**a**) refined mesh and its (**b**) material assignment, demonstrating the finite triangular elements.

**Figure 5 bioengineering-11-00096-f005:**
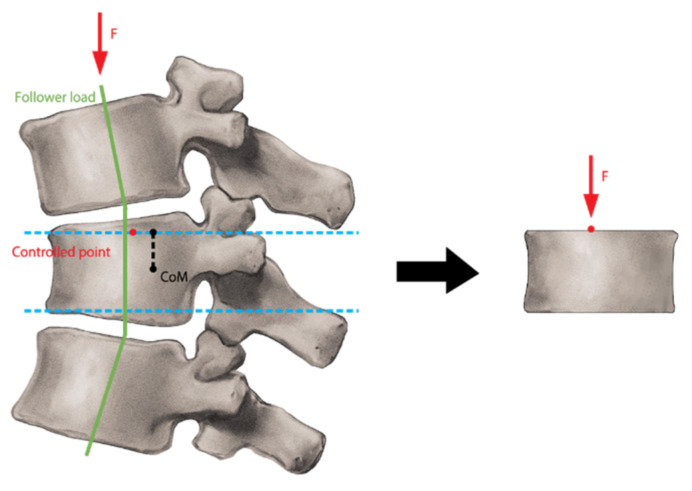
Design of the mechanical test performed on vertebrae demonstrating the follower load and force applied under actual loading anatomy compared to a pure axial compression test of the vertebral body.

**Figure 6 bioengineering-11-00096-f006:**
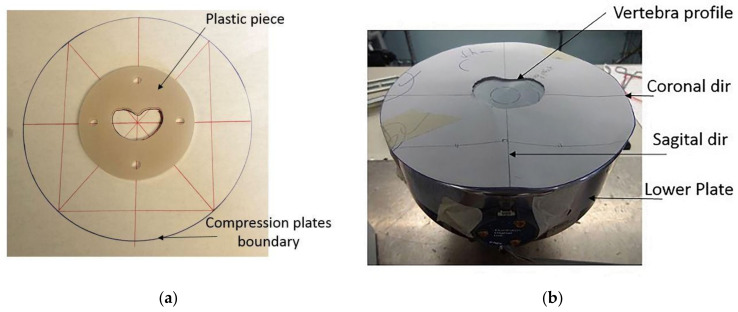
Models to position the samples in the MTS machine. (**a**) A laser-cut, 3D-printed plastic piece oriented on a sheet of paper cut along the vertebral plane was aligned with the (**b**) set up reference points on the MTS machine and used to position the specimen.

**Figure 7 bioengineering-11-00096-f007:**
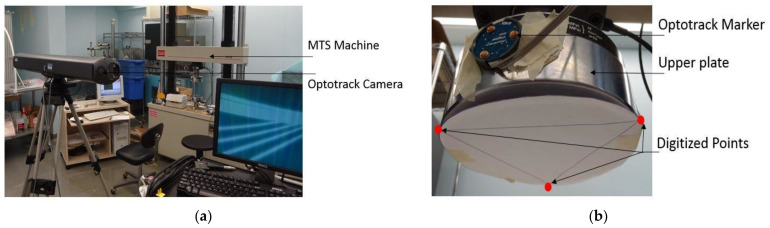
Compression testing setup with the (**a**) MTS compression machine and OPTOTRAK camera equipment. The axial displacement was computed via (**b**) three digitized points coordinates on the upper plate of the MTS machine.

**Figure 8 bioengineering-11-00096-f008:**
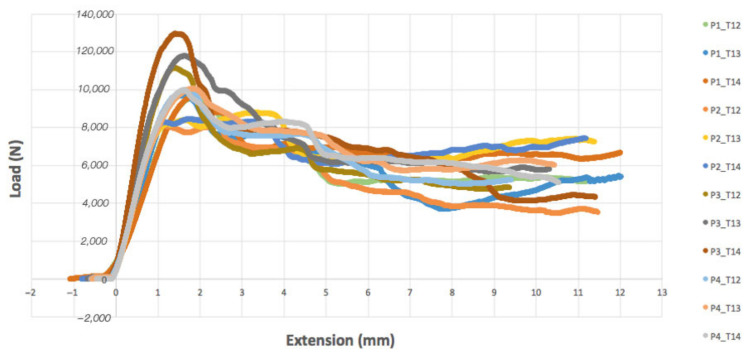
Load–displacement characteristic from experimental thoracic porcine bodies under pure axial compression.

**Figure 9 bioengineering-11-00096-f009:**
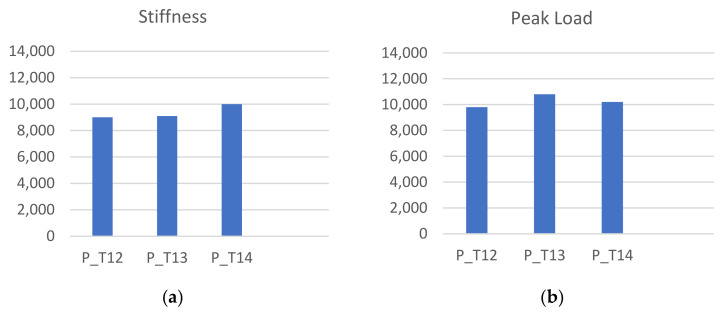
(**a**) Stiffness experimental values by vertebral level and (**b**) peak load experimental values by vertebral level.

**Figure 10 bioengineering-11-00096-f010:**
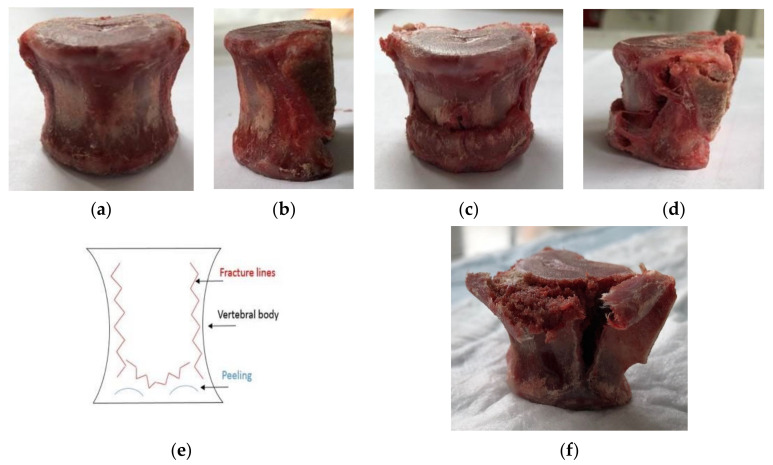
Anterior and lateral views of the (**a**,**b**) pre-compression, intact vertebrae and (**c**,**d**) post-compression, fractured porcine T14 vertebral body, (**e**) explicative scheme showing the failure patterns, and (**f**) superolateral view of fracture pattern, showing splitting of the lateral and anterior surface.

**Figure 11 bioengineering-11-00096-f011:**
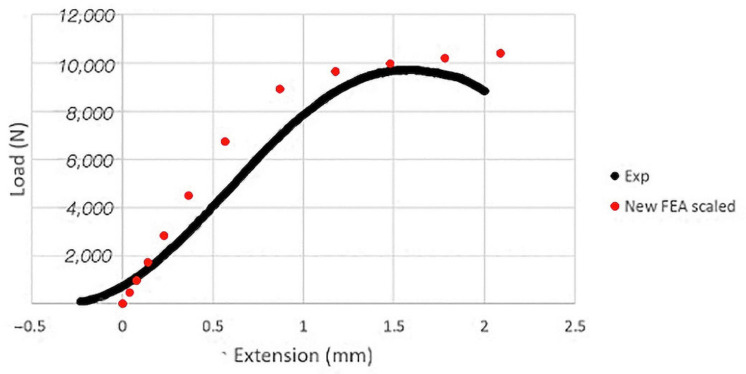
Comparison between the experimental and predicted load–displacement curves for a porcine vertebra after equation adjustment with scaling coefficient.

**Figure 12 bioengineering-11-00096-f012:**
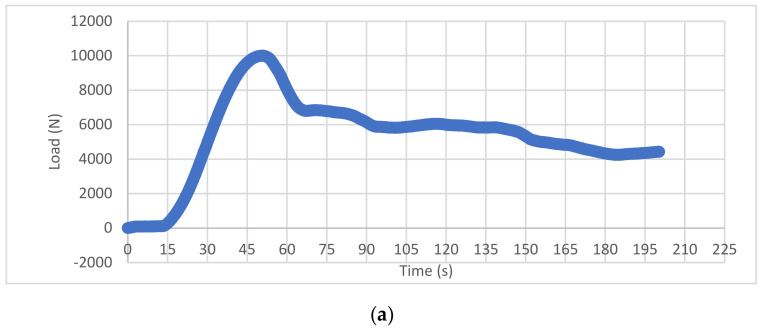
FEM representation of vertebral response to axial load. (**a**) Graph of load per time, where peak load is represented at 55 s, plateau at 120 s, (**b**) mesh of vertebral body at peak load, 55 s, (**c**) 120 s, (**d**) final time with corresponding PEEQ, and (**e**) side-by-side superior view comparing FEM and physical vertebra.

**Table 1 bioengineering-11-00096-t001:** CT scan specifications used in imaging of vertebrae.

Specification	Title 2
X-ray Tube Current	200 mA
Tube peak voltage (kVp)	120 kV
Slice thickness	max = 0.5 mm
Pixel slice	max = 0.27 mm
Slide width	min = 512 px
Slice height	min = 512 px

**Table 2 bioengineering-11-00096-t002:** Calculated constants for porcine vertebral material assignment and FE range.

*a*	*b*	Number FE
0.48	4.6 × 10^4^	162,799–200,758

## Data Availability

Data are contained within the article. The raw data presented in this study are available by contacting the corresponding author.

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
