# Peer review of "Experimentally Validated Finite Element Analysis of Thoracic Spine Compression Fractures in a Porcine Model"

_bioengineering, 2024, doi:10.3390/bioengineering11010096_

Round 1
Reviewer 1 Report
Comments and Suggestions for Authors
Interesting paper about a Finite Element Analysis of thoracic spine compression fractures in a porcine Model. Analysis. The authors used only four porcine spines for their analysis. They may explain , whether this low number of specimen is really statistically sufficient for FEA.
Author Response
Response: Firstly, we thank you for your time and insight in reviewing this manuscript. The concern about the number of spines used for analysis is understandable—however, our work is analogous to existing studies performed in the field, utilizing multiple vertebrae from just a single porcine spine (Brummund et al., 2017) (Bright et al., 2017). We additionally observed consistency across the experimental compression results, which allowed us to perform FEA on multiple vertebrae from one porcine spine.
Reviewer 2 Report
Comments and Suggestions for Authors
Thank you for submitting your paper titled "Experimentally Validated Finite Element Analysis of Thoracic Spine Compression Fractures in a Porcine Model". We are very interested in this field of research and admire your work. After careful review and discussion, we would like to provide you with some suggestions and opinions to help you further improve your paper and make it more consistent with academic standards and publication requirements.
(1) In Section 2.1, we suggest that you further elaborate on why the middle part of the thoracic spine is more prone to wedge compression fractures, and provide relevant research or experimental evidence to support your viewpoint, making your writing more credible and persuasive.
(2) When polishing individual vertebrae, please further explain how to polish, what the purpose of polishing is, and how this affects the experimental results, helping readers better understand your experimental design and the reliability of the results.
(3) On page 4, it mentions the use of MIMICS and 3-MATIC software for three-dimensional reconstruction and refinement of the vertebral body. It is suggested to introduce the principles and applications of these two software, and provide some relevant literature support to help readers understand your experimental process and results.
(4) On page 5, please provide more information about multi-point constraints (MPC), such as how it connects control points with nodes that constitute the skull surface of the vertebral body, and its role and advantages.
(5) For the use of MTS units and OPTOTRAK devices, you can further introduce in detail the working principle, parameter settings, experimental steps of the equipment, as well as the analysis and processing methods of the obtained data.
(6) It is suggested that you introduce in detail the technical route and method flow of experimenting with simulated porcine thoracolumbar wedge fractures and establishing a finite element model based on CT in the method section, and explain the advantages and limitations of these techniques and methods.
(7) It is recommended that you add quality assurance and control measures for data collection and measurement in your study, such as how to ensure the accuracy and stability of the equipment during the experiment, how to eliminate experimental interference and errors, etc.
(8) While it mentions that animal specimens such as pig samples are suitable for similar studies on human spinal finite element models, there is no discussion on the limitations and restrictions of this research method. It is suggested that you discuss this, comparing the differences and applicable scenarios between human cadavers and animal specimens.
Round 2
Reviewer 2 Report
Comments and Suggestions for Authors
My recommendation is accepted as it is.